# Turning Software Engineers into Machine Learning Engineers

**Alexander Schiendorfer** [* 1]   **Carola Gajek** [* 1]   **Wolfgang Reif** [1]

## Abstract

A first challenge in teaching machine learning to software engineering and computer science students consists of changing the methodology from a constructive design-first perspective to an empirical one, focusing on proper experimental work. On the other hand, students nowadays can make significant progress using existing scripts and powerful (deep) learning frameworks – focusing on established use cases such as vision tasks. To tackle problems in novel application domains, a clean methodological style is indispensable. Additionally, for deep learning, familiarity with gradient dynamics is crucial to understand deeper models. Consequently, we present three exercises that build upon each other to achieve these goals. These exercises are validated experimentally in a master's level course for software engineers.

## 1. Teaching Basic Machine Learning

The current state of the art of and popular interest in machine learning (esp. deep learning) is remarkable. There are numerous blogs, tutorials, online-classes, and code snippets that allow for very fast prototyping and experimentation. Endeavors such as fine-tuning a pre-trained image classification network for a special application domain have become *beginner* tutorials (Chilamkurthy, 2017). While we believe that all of these resources are valuable to get students *motivated* and *excited* about the field, we also noted them as potential obstacles in acquiring *basic machine learning principles* that are needed for practitioners keen on applying ML to their domain of choice.

We noted that especially computer science and software engineering students tended to struggle with the adoption of an *empirical* mindset rather than a *constructive* one. That is, instead of *designing* an ML model in terms of its hyper-parameters (e.g., layers, network architecture, etc.) like one would do for, say, a UML class diagram, we need to rely on a systematic approach to experimental determination. That involves proper protocols of experiments, investigation of the learning rates (on train/val sets), or inspection of gradient developments.

To the novice in ML, it might thus be overwhelming to deal with both new algorithms, models, and techniques as well as whole new development style, i.e., adopting an experimental methodology. What should be treated as a black box? What should be understood in more detail? This can be exacerbated by the teachers' emphasis on the former, due to available material and individual curiosity. For instance, numerical optimizers (that offer interfaces to inject gradients) beyond stochastic gradient descent typically need no reimplementation and can be used off the shelf. Knowledge about gradient dynamics, on the other hand, is particularly valuable in terms of diagnosing vanishing/exploding gradient problems that can help to improve training by, e.g., clipping gradients or adding batch-norm layers. Understanding this core material paves the way for more elaborate models such as autoencoders or GANs.

## 2. Practising Essential Machine Learning Skills

The variety of available material and code for ML applications and especially powerful frameworks allow us to quickly build impressive models. However, to get efficiently meaningful models, it is advisable not to rely completely on libraries and frameworks. We have identified three fundamental skills an ML engineer should acquire for it and propose exercises for software engineering students to enhance them[1]. The first skill is a systematic methodology for hyperparameter tuning, as ML users typically train plenty of models with various architectures in order to find the one (together with its corresponding hyperparameters) best performing the desired task. Secondly, ML engineers should know how to perform proper data splitting and the correct usage of the resulting subsets, as this determines the validity of the model. This is not trivial in practice, for instance, for time series data with recurring patterns of different temporal

---

[*]Equal contribution  [1]Institute for Software & Systems Engineering, University of Augsburg, Augsburg, Germany. Correspondence to: Alexander Schiendorfer <schiendorfer@isse.de>.

*Proceedings of the 35th International Conference on Machine Learning*, Stockholm, Sweden, PMLR 80, 2018. Copyright 2018 by the author(s).

---

[1]The sources of the exercises can be accessed via `https://github.com/isse-augsburg/ecml2020-teach-ml`.

horizons. When applying numerical optimizers for training, gradient problems can occur, especially with advanced or customized algorithms. In order to be able to detect and improve them, it is essential to understand the gradient signals, which we have identified as the third skill. This can be the preparation to work with advanced visualization toolkits like TensorBoard[2] to get more insight into the model's dynamics.

## 2.1. Introducing Hyperparameter Tuning

During their education, software engineers are typically taught a constructive operational mode, i.e. creating well-considered architectural and design models before implementing their software. By contrast, the task of tuning hyperparameters for ML models needs a more experimental approach. While there exist empirical guidelines and research findings that point to promising regions or directions of the hyperparameter space, an ML engineer still has to explore these regions in greater detail. Considering an underfitting neural network as a simple example: it is well known that increasing the complexity of the model by adding more neurons or layers or decreasing the regularization will yield better performance. However, finding the optimal number of neurons and layers requires systematic exploration, e.g. using common automatic hyperparameter optimization algorithms like Grid or Random Search.

This union of methodical and explorative approach for hyperparameter tuning is new to software engineers and an important aspect of the tasks of an ML engineer, so it needs to be practiced. Therefore, we designed an exercise for our students focusing on tuning the probably most important hyperparameter (Goodfellow et al., 2016; Smith, 2017) for an exemplifying simple regression problem trained with gradient descent: the learning rate of the optimization algorithm. We chose linear regression as the first model due to its simplicity as the model itself has no hyperparameters. The data set consists of eight two-dimensional data points $(x, y)$ sampled from an unknown, linear curve, alternatively with or without noise in the $y$-values. We designed a widget visualizing the data points, the current linear model $\hat{y}(x) = w_0 + w_1 x$ trying to fit the samples, the corresponding current values of the weights $w_0$ and $w_1$ and the resulting current loss $L(y, \hat{y})$ of the model, see Figure 1. The linear function can be fitted in three different ways:

1. Adjusting the weights manually using the sliders at the bottom of the widget. This is a good starting point to get familiar with the widget and an intuition of suitable weights of the linear function.

2. Gradually training the weights using gradient descent step by step (button `Gradient step`) or repeatedly

until a self-specified stopping criterion is met (button `Solve by gradient descent`). In this mode, the students can experiment with different values of the learning rate and see the effects on the model. The calculated derivatives of the loss by the weights provide an idea of appropriate values of the learning rate.

3. Determining the optimal weights analytically using the normal equations (button `Solve by normal equations`) based on the pseudo-inverse matrix (Goodfellow et al., 2016). This allows on the one hand to check the correctness of the students' gradient calculations and on the other hand to point out the difference between analytical and numerical solution methods.

The button `Show secret` displays the actual weights of the original, true linear function and can show that not even the analytical normal equations can correctly determine the original weights when considering noisy data.

## 2.2. Manage Proper Data Splitting

However, by now the students only learned to tune a hyperparameter of the optimization algorithm, i.e. the learning rate. To actually tune model hyperparameters, splitting up the data set in training and validation sets is necessary. As commonly known, the training set is used to train various models which are afterward evaluated on the unseen instances of the validation set. After choosing the best model, a second holdout set, the test set, gives an approximation of how well the model will perform on new instances in production. To obtain meaningful and comparable results, the individual subsets must be both representative for the whole data set and not be modified during the tuning process.

The second exercise shall teach the students how to split a data set properly according to meaningfulness and reproducibility and how to choose the best model hyperparameters based on these subsets. Specifically, they have to find the optimal polynomial degree to fit a given non-polynomial function. The instances $(x, y)$ are (for presentation reasons) again two-dimensional and sampled from a trigonometry-based function $y(x) = (x - 1)\sin(x + 2) + \mathcal{N}(0, 1.5)$ with normal distributed noise, as displayed in Figure 2. As the students are already familiar with linear regression, we keep it up for this exercise and add model hyperparameters using polynomial basis functions $[x^2, x^3, ..., x^n]$ to simulate polynomial regression instead of introducing this new model. At first, they need to split up the instances properly (as explained above) into the three sets. Based on the current polynomial degree $k = 1, ..., n$, the originally scalar input $x \in \mathbb{R}$ needs to be extended to a vector $\mathbf{h}(x) = [x, x^2, x^3, ..., x^k]^T \in \mathbb{R}^k$ by adding powers of $x$ as new features such that the vectors can be fed into the

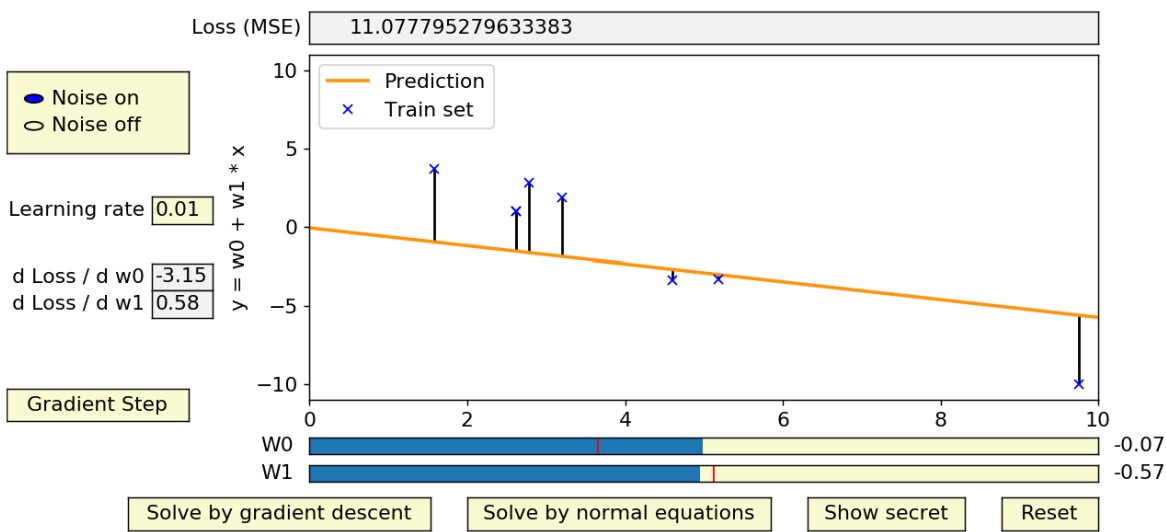

Figure 1. Widget for tuning the learning rate of gradient descent, contains detailed depictions of current gradient and values of $w_0$, $w_1$.

linear regressor $\hat{y}\big(\mathbf{h}(x)\big) = w_0 + w_1 x + w_2 x^2 + ... + w_k x^k$. To choose the best hyperparameter value, i.e. the polynomial degree $k$, the students plot a learning curve displaying the training and validation error for $k = 1, ..., n$. The final task is to calculate the generalization error the model produces on the test set and comparing it to the validation loss.

The exercise has been processed by ten groups of three students, where we could detect that each group had problems with different aspects, both concerning the data handling and the tuning process: Some groups did not take care of the representative segmentation of the data. As a consequence, the validation and test set contained instances from two disjoint areas of the data set, e.g. validation set with $x < 0$ and test set $x \geq 0$. Since the model selection is based on the training and validation set, this segmentation will focus on models well approximating the instances with negative $x$-values and disregard their behavior for non-negative $x$-values. In a few groups, we could observe that the test error was not calculated due to the missing expansion of the original scalar inputs to powers of the input up to degree $k$. The usage of a pre-processing pipeline for the given and incoming instances during production (like applying feature normalization, PCA, etc.) is necessary for complex, real-world applications, too. One group even changed the given code so that the previously fixed segmentation of data was thus performed randomly on each execution which led to non-reproducible results.

Regarding the tuning process, three issues occurred in individual groups: On the one hand, we observed a common error to involve the test set already for selecting the best model, a different group took this decision based only on

the training error - both variants are wrong. The last noticed error concerns the error function that had to be implemented by the students. Since the three data sets are typically not of equal size, the summed error over all instances should be averaged over the size of the respective sets.

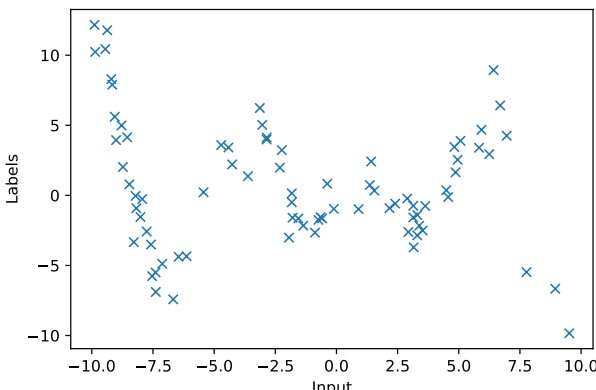

Figure 2. The data set used for the second exercise: 80 samples drawn from the non polynomial function $y = (x - 1)\sin(x + 2) + \mathcal{N}(0, 1.5)$ with normal distributed noise.

### 2.3. Backpropagation and Auto-Differentiation

To train deeper models than linear regression, the automatic derivation of gradients of the loss function with respect to each weight is a key element. Backpropagation, as introduced for classical multilayer perceptrons (Rumelhart et al., 1985), is the algorithm that runs deep learning. Approaching it for the first time might however feel daunting. In our experience, including many remarks about reverse-mode autodiff,

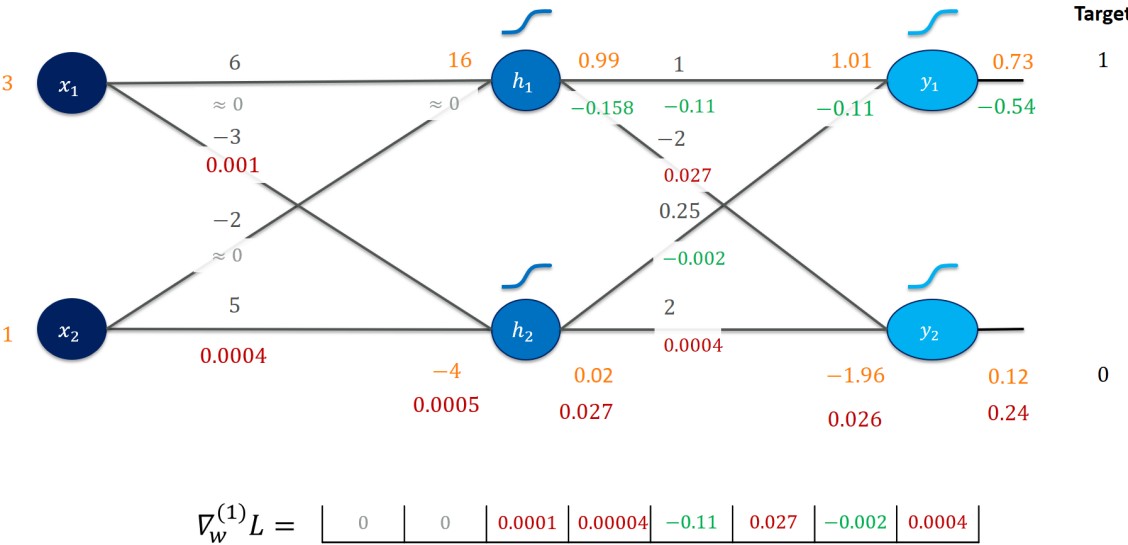

$$\nabla_w^{(1)} L = \begin{array}{|c|c|c|c|c|c|c|c|} \hline 0 & 0 & 0.0001 & 0.00004 & -0.11 & 0.027 & -0.002 & 0.0004 \\ \hline \end{array}$$

*Figure 3.* A fully worked example of a 2-2-2 neural networks with current weights/activations in gray/orange.

chain-rule, Jacobian, or delta-rule will have students either ignoring the material ("the autograd framework will do the job") or spending a generous amount of paper deriving equations by hand. Yet a healthy understanding of the gradients' purpose is critical. Recent textbooks (e.g., (Géron, 2017)) have begun to include discussions on automatic differentiation on simpler functions such as $f(x, y) = x^2 + 2xy + 5$ as one ingredient of backpropagation (the other one being the actual parameter update, i.e., the gradient step). However, that approach might feel discontinuous to students, leaving unclear what parts of the gradients of a neural network's parameters are derived automatically. For example, students associated autodiff with the activation functions alone and thought of backpropagation as an encompassing algorithm.

Therefore, our remedy was to work through the simplest multidimensional neural network imaginable, i.e., a 2-2-2 network as depicted in Figure 3 with a simple sum of squared errors as loss. Having two outputs is beneficial since, e.g., the output of $h_2$ affects both $y_1$ and $y_2$, immediately requiring the multivariate chain rule. In that example, $y_1$ should go up and $y_2$ should simultaneously go down. Having a numerical example allows for "interpreting the gradients". For instance, considering the weight connecting $h_1$ and $y_2$ (currently $-2$), students can verify that a small increase from $-2$ to $-1.9$ would increase the input to (and consequently the output of) $y_2$, which is what we would want to avoid.

The example network can be displayed when going through backpropagation step-by-step, interleaved with the formal steps necessary to calculate the involved gradients (back from the loss, back through an activation, back to the

weights of a layer, etc.). The particular example we selected also highlights problems with vanishing gradients that emerge when using sigmoidal activation functions that are improperly initialized (i.e., much too high in this case). Later, this example is also useful to demonstrate how batch normalization mitigates these issues and retains much more usable gradient signals in earlier layers even with sigmoid.

In terms of practical exercises, the network can be used simply to have students calculate (some of) the gradients manually, along with "checkpoints" such as that $\frac{\partial L}{\partial w_{1,1}}$ should be $-0.11$. But it is probably more sensible to have them code it in a programming language of their choice and keep it extensible (try out different activation functions, write a tiny autograd library, etc). In addition, we asked them to interpret gradients the aforementioned way for different training instances.

## 3. Conclusion

We reflected on our experiences with teaching essential machine learning skills to software engineering students, emphasizing the shift from a constructive to an empirical mindset and the temptations of existing powerful black box code snippets that need to be backed up by proper grasp of the fundamentals. Based on what we assume to be the more frequently recurring tasks for ongoing machine learning engineers, we suggested a lean teaching concept that focuses on experimentation and understanding of gradient dynamics. For all our proposed exercises, Jupyter notebooks are made available and we hope to initiate a collection of interesting code examples to turn software engineers into ML engineers.

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
