# OpenReview forum: "Turning Software Engineers into Machine Learning Engineers"
_ECMLPKDD.org/2020/Workshop/TeachML — ECML PKDD 2020 TeachML_

### Official Review · AnonReviewer3 · 2020-07-18
**Self-explanatory materials for proper understanding of ML basics.**

**Rating:** 9
**Confidence:** 5

**Review:**

### Summary:
This work focuses on teaching Machine Learning (ML) to Software Engineers (SE). There is a need for change of thought process between SE and ML due to constructive and empirical mindset. Authors specifically focused on three aspects of ML, namely, 1. Hyperparam tuning, 2. Data handling and 3. Backprop. To address these challenges, authors created materials which explains the underlying concepts and how they differ from SE. Authors report various findings on the effectiveness of the materials and teaching methodology on the group of students.

### Strong Points:
* Materials are self-explanatory and useful for all students over and beyond SE students
* Motivation on each material is well thought out and described very succinctly

### Weak Points:
* I found the description on the 3rd material is a bit lacking on the self-explanation part compared to the other two materials, although the code is written well with sufficient comments in all
* A little more detail on the “normal equation” might be helpful for comparison to students who are not aware of it

### Other comments:
* Small suggestion: Resolution of Fig. 2 is very low. Generating the figures in either “pdf” or “eps” format might retain the figure quality

---

### Official Review · AnonReviewer1 · 2020-07-27
**Evaluation of  ML teaching principles using jupyter notebooks for software engineers**

**Rating:** 7
**Confidence:** 3

**Review:**

The authors of this submission identified three major attention points of ML teaching when introducing the topic to software engineering students. These are hyperparameter tuning,
proper data splitting and knowledge of gradient computations and workings of backpropagation.

Those principles are taught with the help of self created jupyter notebooks, providing an interactive widget for a simple linear regression problem for visualizing the effect of different learning rates, the proper use of data splitting for hyperparameter tuning and an example implementation of automatic differentiation.

What I really liked about the paper is that the authors reported first hand experiences of their students using this approach, including what went wrong in many cases. Unsurprisingly, this seemed to have been the part about proper data splitting. This identifies areas which probably need to be addressed better in future design of teaching materials.
Also, the authors invested great efforts in the very readable design of small toy classes showing how backpropagation via autodiff can be implemented. I assume this will be of great value for understanding the working principles of large frameworks like pytorch or tensorflow, especially for software engineering students.

On the other hand, I think that such a code-heavy approach might not be directly transferable to different audiences which might be less familiar with programming or software design.

Moreover, even though I agree that using jupyter notebooks has become a frequently used and pretty much standard way for teaching and demonstration within the data science community, I doubt that the authors used it as their only means of teaching in ML classes and they could have elaborated a bit on how they encapsulate their notebooks into their regular teaching. But maybe that had to be left out due to paper space restrictions.

---

### Official Review · AnonReviewer2 · 2020-07-29
**Hands-on exercises for SWEs to learn ML**

**Rating:** 6
**Confidence:** 3

**Review:**

Summary:
The paper first talks about the current state of learning machine learning and the challenges software engineers face when learning machine learning. The authors suggest hyperparameter tuning,  data splitting, and gradient signals are the main topics to learn.

Strong Points:
1. Authors designed specific exercises to help students learn hyperparameter tuning, data splitting, and gradient signals.
2. Specific exercises can be used not just for software engineers, but for anyone with an engineering or quantitative background.
3. Explanations of the exercises and what they provide to students are clear.

Weak Points:
1. The entire paper is based on the observation "We noted that especially computer science and software engineering students tended to struggle with the adoption of an empirical mindset rather than a constructive one." While it sounds reasonable, there are no additional examples or details explaining how software engineers struggle or why.
2. It's not clear how the authors chose hyperparameter tuning,  data splitting, and gradient signals as the main topics. Why are these more important than learning about model architectures and their applications?
3. There's no quantitative difference provided showing the effectiveness of the results provided. I would have liked to see pre-test and post-test scores of students based on this teaching method. It would have been even nicer if there was a way to compare it to existing teaching methods.

---

### Decision · Program_Chairs · 2020-07-31

**Decision:**

Accept

**Comment:**

The reviewers agree that this paper will be accepted. Thank you for your contributions.

Please register with the conference as soon as possible! See this page for details:
https://ecmlpkdd2020.net/attending/registration/.
Which asks that at least one author per paper registers until July 31, 2020.
We apologize for the very short notice.